# 🧙 *WizardCoder*: Empowering Code Large Language Models with Evol-Instruct

**Ziyang Luo**[2*]  **Can Xu**[1*]  **Pu Zhao**[1]  **Qingfeng Sun**[1]  **Xiubo Geng**[1]
**Wenxiang Hu**[1]  **Chongyang Tao**[2]  **Jing Ma**[2†]  **Qingwei Lin**[1]  **Daxin Jiang**[1†]

[1]Microsoft
[2]Hong Kong Baptist University
```
{cszyluo, majing}@comp.hkbu.edu.hk, {caxu,puzhao}@microsoft.com
{qins,xigeng,wenxh,chongyang.tao,qlin,djiang}@microsoft.com
```

## Abstract

Code Large Language Models (Code LLMs), such as StarCoder, have demonstrated remarkable performance in various code-related tasks. However, different from their counterparts in the general language modeling field, the technique of instruction fine-tuning remains relatively under-researched in this domain. In this paper, we present *Code Evol-Instruct*, a novel approach that adapts the *Evol-Instruct* method to the realm of code, enhancing Code LLMs to create novel models *WizardCoder*. Through comprehensive experiments on five prominent code generation benchmarks, namely HumanEval, HumanEval+, MBPP, DS-1000, and MultiPL-E, our models showcase outstanding performance. They consistently outperform all other open-source Code LLMs by a significant margin. Remarkably, *WizardCoder 15B* even surpasses the well-known closed-source LLMs, including Anthropic's Claude and Google's Bard, on the HumanEval and HumanEval+ benchmarks. Additionally, *WizardCoder 34B* not only achieves a HumanEval score comparable to GPT3.5 (ChatGPT) but also surpasses it on the HumanEval+ benchmark. Furthermore, our preliminary exploration highlights the pivotal role of instruction complexity in achieving exceptional coding performance.

## 1 Introduction

Recently, Large Language Models (LLMs) (Brown et al., 2020; OpenAI, 2023; Chowdhery et al., 2022; Anil et al., 2023; Hoffmann et al., 2022; Rae et al., 2021; Zeng et al., 2022; Zhang et al., 2022; Touvron et al., 2023a) have garnered immense attention and demonstrated impressive success. Notably, OpenAI's GPT3.5 (ChatGPT) stands out as a prominent example. These models, through extensive pre-training on vast internet data and fine-tuning with detailed instruction data (Ouyang et al., 2022), have achieved state-of-the-art (SOTA) zero-shot performance across diverse NLP tasks. This trend also extends to the realm of code understanding and generation, where a multitude of Code LLMs have emerged (Chen et al., 2021a; Li et al., 2022; Fried et al., 2022; Nijkamp et al., 2023b; Zheng et al., 2023; Wang et al., 2021; 2023; Li et al., 2023a; Nijkamp et al., 2023a; Rozière et al., 2023). These models, pre-trained on substantial code data, excel in various code-related tasks, consistently delivering impressive performance.

In contrast to most previous Code LLMs that primarily focus on the pre-training process, there has been limited exploration of fine-grained instruction tuning in the code domain. The introduction of instruction tuning was initially designed to enhance the generalization capabilities of LMs across different tasks via multitask training (Raffel et al., 2020; Wei et al., 2022; Chung et al., 2022; Aribandi et al., 2022; Sanh et al., 2022; Xu et al., 2022; Khashabi et al., 2020). OpenAI's InstructGPT (Ouyang et al., 2022), for instance, involved soliciting human annotators to provide explicit instructions to ensure alignment with users' intentions. Similarly, recent works such as Alpaca (Taori et al., 2023) employed the self-instruct (Wang et al., 2022) method, where GPT3.5 (ChatGPT) generated the

---

* Equal contribution. Work done during the internship at Microsoft.
† Corresponding author.

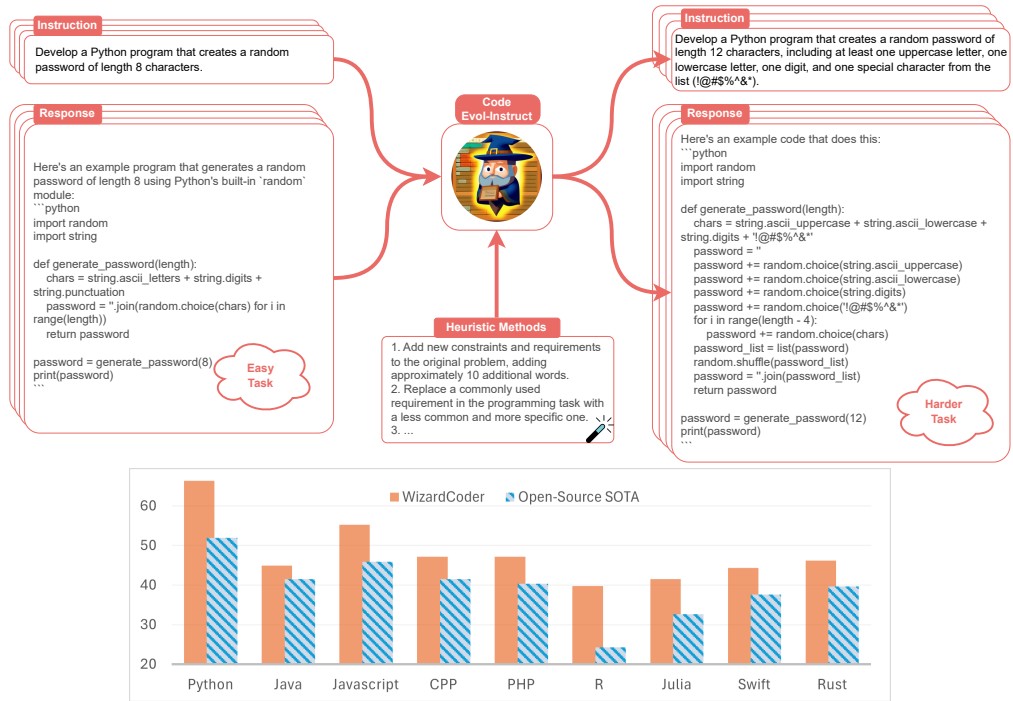

Figure 1: An illustration of our novel *Code Evol-Instruct* and the superior pass@1 performance of our *WizardCoder 34B*, outperforming the open-source SOTA (CodeLlama-34B-Series as of the date before August 24, 2023) by a large margin in 9 different programming languages. The Python score is the mean between HumanEval and MBPP.

instruction data. Vicuna (Chiang et al., 2023) utilized user-shared conversations collected from ShareGPT.com. WizardLM (Xu et al., 2023) introduces the *Evol-Instruct* method, which involves evolving existing general instruction data to generate more complex and diverse datasets. Drawing inspiration from these previous works in the general domain, our work, *Code Evol-Instruct*, is specifically tailored to the coding domain's distinctive characteristics.

In this study, we aim to enhance the capabilities of the SOTA open-source Code LLMs (i.e., StarCoder and CodeLlama), by introducing our novel *Code Evol-Instruct*. The motivation of this fine-grained instruction-tuning method in the code domain is to automatically increase the complexity of code instruction data, so as to make the best of the internal coding ability of the Code LLMs. Our *Code Evol-Instruct* incorporates several novel methods, including heuristics tailored to coding task features, adversarial sample heuristics, time/space complexity requirements, and evolving stop controls. The whole process includes two steps: initially, we apply our *Code Evol-Instruct* to evolve basic code instruction data, specifically Code Alpaca (Chaudhary, 2023). Subsequently, we fine-tune StarCoder and CodeLlama using our newly generated code instruction-following training set, resulting in our *WizardCoder* models.

Figure 1 and the experimental results obtained from five code generation benchmarks, namely HumanEval (Chen et al., 2021b), HumanEval+ (Liu et al., 2023), MBPP (Austin et al., 2021), DS-100 (Lai et al., 2022), and MultiPL-E (Cassano et al., 2022), demonstrate that our *WizardCoder* models outperform all other open-source Code LLMs (before August 24, 2023), achieving state-of-the-art (SOTA) performance. Remarkably, our *WizardCoder 15B* even surpasses well-known Anthropic's Claude and Google's Bard in terms of pass rates on HumanEval and HumanEval+. Furthermore, *WizardCoder 34B* not only achieves a HumanEval score comparable to GPT3.5 (ChatGPT) but also surpasses it on the HumanEval+ benchmark. Beyond this, our preliminary studies indicate that the complexity of instructions is the key to achieving exceptional coding performance.

The contributions of this work can be summarized as follows:

- We introduce *Code Evol-Instruct*, a novel instruction fine-tuning approach for code, which enhances the performance of the open-source Code LLMs by a large margin.

- We develop *WizardCoder* models, which surpass all other open-source Code LLMs by a substantial margin in coding tasks. Notably, the 15B version even outperforms the well-known closed-source LLMs, such as Claude, and Bard. The 34B version achieves a HumanEval score comparable to GPT3.5 (ChatGPT) and surpasses it on the HumanEval+ benchmark.

- We conduct a preliminary study highlighting the pivotal role of instruction complexity in achieving exceptional coding performance.

## 2 RELATED WORK

**Large Language Models.** Recently, LLMs have demonstrated remarkable achievements across a broad spectrum of tasks. Prominent tech companies have made significant strides in developing highly proficient LLMs. These include OpenAI's GPT3&4 (Brown et al., 2020; OpenAI, 2023), Google's PaLM (Chowdhery et al., 2022; Anil et al., 2023), and Bard[1], DeepMind's Chinchilla (Hoffmann et al., 2022), and Gopher (Rae et al., 2021), as well as Anthropic's Claude[2]. However, it is important to note that these models are closed-source and can only be accessed through specific APIs or may not be accessible at all.

The AI community has witnessed the release of several open-source LLMs, where the model weights are made publicly available. EleutherAI has contributed GPT-NeoX-20B (Black et al., 2022) and GPT-J-6B (Wang & Komatsuzaki, 2021). Google has released UL2-20B (Tay et al., 2022). Tsinghua University has introduced GLM-130B (Zeng et al., 2022). Meta has released OPT (Zhang et al., 2022) and LLaMA1&2 (Touvron et al., 2023a;b). It is worth noting that while these open-source models have made valuable contributions, they generally do not exhibit the same level of performance as their closed-source counterparts.

**Large Language Models for Code.** Recent studies have introduced a significant number of LLMs for code-related tasks to address the challenges of code understanding and generation. OpenAI has unveiled Codex (Chen et al., 2021a) and Code-Davinci (Microsoft, 2023). Google has proposed PaLM-Coder (Chowdhery et al., 2022). They perform outstandingly on the popular code completion benchmarks, like HumanEval (Chen et al., 2021b) and MBPP (Austin et al., 2021). However, these models are closed-source.

On the other hand, there are several open-source Code LLMs available. Salesforce has introduced CodeGen1&2 (Nijkamp et al., 2023b;a), CodeT5 (Wang et al., 2021), and CodeT5+ (Wang et al., 2023). Tsinghua University has contributed CodeGeeX (Zheng et al., 2023), and the BigCode Project has developed StarCoder (Li et al., 2023a). Meta has released the CodeLlama-Series (Rozière et al., 2023), which achieves open-source SOTA performance on several benchmarks. The closely related model, CodeLlama-Instruct, refines its performance through the self-instruct method. These models have demonstrated notable advancements in code-related tasks. However, when compared to the SOTA closed-source models, they still lag behind significantly. In contrast to the aforementioned models, our work demonstrates that further training Code LLMs with our *Code Evol-Instruct* can substantially enhance performance.

**Instruction Fine-Tuning.** The primary objective of instruction fine-tuning in its early stages was to enhance the cross-task generalization capabilities of LMs. This was achieved by fine-tuning LMs with a substantial corpus of public NLP tasks. T5 (Raffel et al., 2020) was among the first models to explore this approach, training on a multitude of supervised text-to-text tasks. Subsequent works such as FLAN (Wei et al., 2022), ExT5 (Aribandi et al., 2022), T0 (Sanh et al., 2022), and UnifiedQA (Khashabi et al., 2020) further expanded the range of tasks to bolster the overall generalization ability of LMs. Notably, ZeroPrompt (Xu et al., 2022) and FLAN-T5 (Chung et al., 2022) pushed the envelope by incorporating thousands of tasks in their training pipelines. Across these studies, a consistent finding emerges: fine-tuning LMs with diverse NLP task instructions yields significant performance improvements when applied to new tasks.

---

[1] https://bard.google.com/
[2] https://www.anthropic.com/index/introducing-claude

While fine-tuning LMs with diverse NLP tasks has shown promising results, it often falls short in aligning with the intentions of real-world users. OpenAI has pursued a different approach by soliciting human annotators to provide a large corpus of human instructions, encompassing diverse forms and a wide range of task types. Building upon this dataset, OpenAI trained its GPT3 (Brown et al., 2020) model to create InstructGPT (Ouyang et al., 2022), which better aligns with users' inputs. This line of development has even led to the impressive work known as GPT3.5 (ChatGPT). However, it is important to note that the dataset and model weights associated with these advancements are not publicly available. Alpaca (Taori et al., 2023) takes a different route by adopting the self-instruct method (Wang et al., 2022), leveraging GPT3.5 (ChatGPT) to generate data for training. Vicuna (Chiang et al., 2023) utilizes user-shared conversations collected from ShareGPT.com to train its models. WizardLM (Xu et al., 2023) introduces the *Evol-Instruct* method, which involves evolving existing general instruction data to generate more complex and diverse datasets. Drawing inspiration from this idea, our work, *Code Evol-Instruct*, aligning with the distinctive characteristics of coding domains, is the first instruction fine-tuning method explicitly designed to enhance Code LLMs.

## 3 WIZARDCODER: SOTA OPEN-SOURCE CODE LLM

In this section, we elaborate on the methodological details of *WizardCoder*. As illustrated in Figure 1, we first adopt our *Code Evol-Instruct* to iteratively evolve the Code Alpaca dataset. Subsequently, we fine-tune the pre-trained Code LLMs with the evolved data.

### 3.1 CODE EVOL-INSTRUCT

Inspired by the *Evol-Instruct* method proposed by WizardLM Xu et al. (2023), this work attempts to automatically enhance the complexity of code instructions, thereby improving the fine-tuning effectiveness of Code LLMs. Diverging from the general domain, our methods are meticulously designed to align with the specific characteristics of coding domains. The evolutionary process introduces the following features:

1. Heuristics aligned with coding task features on platforms like LeetCode, strategically increasing the complexity of coding tasks to enhance the model's capabilities.

2. Introduction of erroneous code as an adversarial sample, inspired by prior research on attacking pre-trained code models Yang et al. (2022); Jha & Reddy (2022), adds a novel and effective method to escalate task complexity.

3. Introduction of a heuristic emphasizing time and space complexity leverages insights from previous studies Madaan et al. (2023), providing a valuable avenue for improving task complexity.

So, the code evolutionary prompt template is as follows:

---

**Prompt for Code Evol-Instruct**

Please increase the difficulty of the given programming test question a bit.

You can increase the difficulty using, but not limited to, the following methods:
{method}

{question}

---

Here, {question} represents the current code instruction awaiting evolution, and {method} is the type of evolution. The five types we used are listed as follows:

---

**Code Evolution Heuristic Methods**

Add new constraints and requirements to the original problem, adding approximately 10 additional words.

Replace a commonly used requirement in the programming task with a less common and more specific one.

If the original problem can be solved with only a few logical steps, please add more reasoning steps.

Provide a piece of erroneous code as a reference to increase misdirection.

Propose higher time or space complexity requirements, but please refrain from doing so frequently.

---

## 3.2 TRAINING *WizardCoder*

We employ the following procedure to train *WizardCoder*. Initially, we utilize StarCoder 15B (Li et al., 2023a) and CodeLlama-34B-Python (Rozière et al., 2023) as the foundations and proceed to fine-tune them using the code instruction-following training set, which was evolved through *Code Evol-Instruct*. The prompt format for fine-tuning is outlined as follows:

---

**Prompt for Fine-Tuning Format**

Below is an instruction that describes a task, paired with an input that provides further context. Write a response that appropriately completes the request.

### Instruction:
{instruction}

### Response:

---

To construct the training dataset, we initialized it with the instruction-following dataset called Code Alpaca[3]. We iteratively employ the *Code Evol-Instruct* technique on this dataset consisting of around 20k samples to produce evolved data. After each round of data evolution, we merge the evolved data from all previous rounds with the original dataset to finetune Code LLMs. An external dev set serves as the controlled Evol Stop. If the performance drops, we halt the evolution. In Appendix C, we outline the approach employed to prevent data leakage. Additionally, Appendix D showcases some evolved examples for reference.

## 4 EXPERIMENT

This section begins by providing a comprehensive overview of the baseline models in our experiments. Subsequently, we present the performance of our models on five code generation benchmarks: HumanEval (Chen et al., 2021b), HumanEval+ (Liu et al., 2023), MBPP (Austin et al., 2021), DS-1000 (Lai et al., 2022) and MultiPL-E (Cassano et al., 2022).

## 4.1 BASELINES

**Closed-Source Models.** Multiple technology companies have successfully developed highly proficient LLMs while choosing not to publicly release them. These models are referred to as closed-source models. For our research, we incorporate a substantial number of these models as our baselines. Specifically, our baselines encompass the following: (i) OpenAI's GPT3.5(ChatGPT)&GPT4 (OpenAI, 2023), Code-Davinci-002 (Microsoft, 2023), Code-Cushman-001 (Microsoft, 2023), and

---

[3]https://github.com/sahil280114/codealpaca

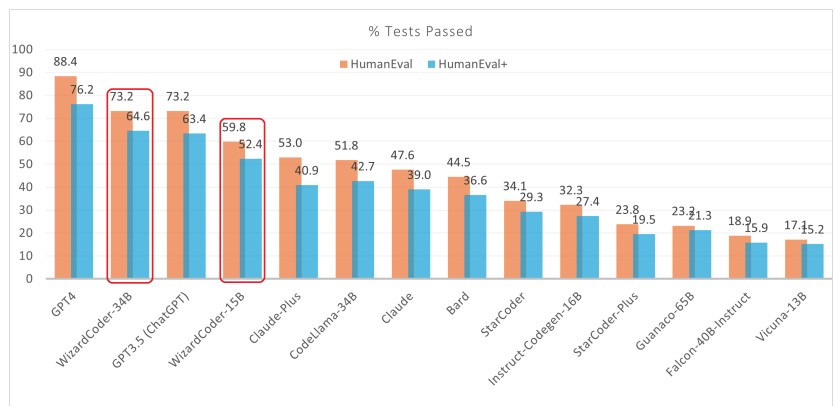

Figure 2: The percentage of pass rates on the HumanEval and HumanEval+ with a single attempt (greedy decoding), following the EvalPlus leaderboard (Liu et al., 2023).

Codex (Chen et al., 2021a); (ii) Google's Bard, PaLM 2 (Anil et al., 2023), PaLM (Chowdhery et al., 2022), and LaMDA (Thoppilan et al., 2022); (iii) Google DeepMind's AlphaCode (Li et al., 2022);(iv) Anthropic's Claude; (v) Huawei's PanguCoder2 (Shen et al., 2023); and (vi) Meta's Unnatural-CodeLlama-34B (Rozière et al., 2023).

**Open-Source Models.** Several open-source LLMs (OSS) have been made available to the AI community, although their performance generally lags behind the closed-source models a lot. As part of our research, we incorporate a significant number of these open-source models as our baselines. Our baselines encompass the following models: InCoderFried et al. (2022), StarCoder and StarCoder-Plus (Li et al., 2023a), LLaMa1&2 (Touvron et al., 2023a;b), CodeGen (Nijkamp et al., 2023b), CodeGeeX (Zheng et al., 2023), CodeT5+(Wang et al., 2023), and CodeLlama (Rozière et al., 2023). In addition, we also include several models with instructions fine-tuning, including CodeLlama-Instruct (Rozière et al., 2023), OctoCoder (Muennighoff et al., 2023), InstructCodeT5+ (Wang et al., 2023), Instruct-Codegen-16B,[4] Guanaco-65B (Dettmers et al., 2023), Falcon-40B-Instruct (Penedo et al., 2023) and Vicuna-13B (Chiang et al., 2023). More details can be found in the Appendix B.

## 4.2 IMPLEMENTATION DETAILS

The StarCoder and CodeLlama-34B-Python serve as our basic foundation models. OpenAI's gpt3.5-turbo is used to evolve the dataset and generate responses. The evolved dataset consists of approximately 78k samples. To fine-tune the basic models, we employ specific configurations, including a batch size of 512, a sequence length of 2048, 200 fine-tuning steps, 30 warmup steps, a learning rate of 2e-5, a Cosine learning rate scheduler, and fp16 mixed precision.

## 4.3 EVALUATION ON HUMANEVAL, HUMANEVAL+, AND MBPP

HumanEval (Chen et al., 2021b), HumanEval+ (Liu et al., 2023), and MBPP (Austin et al., 2021) are key benchmarks in the Code LLM field, featuring diverse Python programming problems validated using test cases. HumanEval comprises 164 problems with an average of 9.6 test cases per problem. HumanEval+ expands the test cases significantly to an average of 774.8 per problem. In contrast, MBPP provides 500 test programming problems with three automated test cases each.[5]

**Comparing with the Closed-Source Models.** Following the same setting of the EvalPlus leaderboard (Liu et al., 2023). In Figure 2, we compare our *WizardCoder* models with the closed-source models, such as GPT4, Claude, and Bard on this leaderboard. Notably, all models generate code

---

[4]https://huggingface.co/sahil2801/instruct-codegen-16B

[5]For a fair comparison, we present results for GPT3.5(ChatGPT)&GPT4 using Eval-Plus with the latest OpenAI's APIs (Liu et al., 2023) (Figure 2) and OpenAI's report (OpenAI, 2023) (Table 1). Prompt format details are in Appendix A.

Table 1: Results of pass@1(%) on HumanEval and MBPP. We follow the previous works (Chen et al., 2021b) to generate n=200 samples to estimate the pass@1 score of our *WizardCoder* models with the same set of hyper-parameters: temperate=0.2, and top_p=0.95. *: our reproduced results.

| Model | Params | HumanEval | MBPP |
|---|---|---|---|
| Closed-source models | | | |
| LaMDA (Thoppilan et al., 2022) | 137B | 14.0 | - |
| AlphaCode (Li et al., 2022) | 1.1B | 17.1 | - |
| PaLM (Chowdhery et al., 2022) | 540B | 26.2 | 36.8 |
| PaLM-Coder (Chowdhery et al., 2022) | 540B | 36.0 | 47.0 |
| PaLM 2-S (Anil et al., 2023) | Unknown | 37.6 | 50.0 |
| Codex (Chen et al., 2021a) | 2.5B | 21.4 | - |
| Codex (Chen et al., 2021a) | 12B | 28.8 | - |
| Code-Cushman-001 (Microsoft, 2023) | Unknown | 33.5 | 45.9 |
| Code-Davinci-002 (Microsoft, 2023) | Unknown | 47.0 | 58.1 |
| GPT-3.5 (ChatGPT) (OpenAI, 2023) | Unknown | 48.1 | 52.2 |
| PanguCoder2 (Shen et al., 2023) | 15B | 61.6 | - |
| Unnatural-CodeLlama (Rozière et al., 2023) | 34B | 62.2 | 61.2 |
| GPT-4 (OpenAI, 2023) | Unknown | 67.0 | - |
| Open-source models | | | |
| Llama (Touvron et al., 2023a) | 65B | 23.7 | 37.7 |
| Llama2 (Touvron et al., 2023b) | 70B | 29.9 | 45.0 |
| CodeGen-Mono (Nijkamp et al., 2023b) | 16B | 29.3 | 35.3 |
| CodeGeeX (Zheng et al., 2023) | 13B | 22.9 | 24.4 |
| StarCoder (Li et al., 2023a) | 15B | 33.6 | 43.6* |
| CodeT5+ (Wang et al., 2023) | 16B | 30.9 | - |
| InstructCodeT5+ (Wang et al., 2023) | 16B | 35.0 | - |
| OctoCoder (Muennighoff et al., 2023) | 15B | 46.2 | - |
| CodeLlama (Rozière et al., 2023) | 34B | 48.8 | 55.0 |
| CodeLlama-Python (Rozière et al., 2023) | 34B | 53.7 | 56.2 |
| CodeLlama-Instruct (Rozière et al., 2023) | 34B | 41.5 | 57.0 |
| *WizardCoder* | 15B | **57.3** | **51.8** |
| *WizardCoder* | 34B | **71.5** | **61.2** |

solutions for each problem utilizing a single attempt, and the resulting pass rate percentage is reported. To maintain consistency, we employ the same experimental setup by generating answers using greedy decoding and evaluate our *WizardCoder* models using the provided evaluation codes.

As depicted in Figure 2, our *WizardCoder 34B* attains the second position in this benchmark, surpassing GPT3.5 (ChatGPT, 64.6 vs. 63.4) on HumanEval+. Our 15B version outperforms Claude-Plus (59.8 vs. 53.0) and Bard (59.8 vs. 44.5). Furthermore, our *WizardCoder* models demonstrate a remarkable superiority over other open-source LLMs that undergo instruction fine-tuning.

**Comparing with the Open-Source Models.**   In Table 1, we conduct a comprehensive comparison of our *WizardCoder* with other open-source models on the HumanEval and MBPP benchmarks. In contrast to the results presented in Figure 2, we adhere to the approach outlined in previous studies Chen et al. (2021b) by generating n samples for each problem to estimate the pass@1 score. The findings presented in Table 1 clearly demonstrate that our *WizardCoder* exhibits a substantial performance advantage over all the open-source models.

## 4.4 EVALUATION ON MULTI-LANGUAGE CODING

We included comprehensive assessment results across 8 distinct programming languages on the MultiPL-E benchmarks. These languages encompass Java, JavaScript, C++, PHP, R, Julia, Swift, and Rust. The empirical results, as presented in Table 2, distinctly demonstrate the superior performance of our *WizardCoder* models across all evaluated programming languages, surpassing the SOTA open-source Code LLMs. This underscores the efficacy of our *Code Evol-Instruct* method.

## 4.5 EVALUATION ON DS-1000

The DS-1000 benchmark Lai et al. (2022) comprises 1k distinct data science workflows spanning 7 libraries. It assesses the performance of code generations against test cases and supports two evaluation modes: completion and insertion. In our experiments, we only report insertion scores for

models that support. In Table 3, we present pass@1 (n=40) results for each library, along with an overall score.[6] Based on these results, our conclusion is that *WizardCoder* demonstrates a significant superiority over all other models when tackling data science problems on the DS-1000 benchmark.

Table 2: Results of pass@1(%) on 8 different programming languages on the MultiPL-E (Cassano et al., 2022) benchmarks. All models are evaluated with the same set of hyper-parameters: temperature=0.2, top_p=0.95, max_length=512, and n=50.

| Model | Params | Java | Js | CPP | PHP | R | Julia | Swift | Rust |
|---|---|---|---|---|---|---|---|---|---|
| CodeGen-Multi | 16B | 22.2 | 19.2 | 21.0 | 8.4 | 6.5 | 0 | 1.3 | 4.2 |
| CodeGeeX | 13B | 19.1 | 16.9 | 16.9 | 13.5 | 3.9 | 0.3 | 7.3 | 7.9 |
| Code-Cushman-001 | - | 31.9 | 31.3 | 30.6 | 29.0 | 11.0 | 1.5 | 22.1 | 25.2 |
| StarCoderBase | 15B | 28.5 | 31.7 | 30.6 | 26.8 | 10.2 | 21.1 | 16.7 | 24.5 |
| StarCoder | 15B | 30.2 | 30.8 | 31.6 | 26.1 | 15.5 | 23.0 | 22.7 | 21.8 |
| CodeLlama | 34B | 40.2 | 41.7 | 41.4 | 40.4 | 22.7 | 31.4 | 35.3 | 38.7 |
| CodeLlama-Python | 34B | 39.5 | 44.7 | 39.1 | 39.8 | 22.4 | 31.4 | 34.3 | 39.7 |
| CodeLlama-Instruct | 34B | 41.5 | 45.9 | 41.5 | 37.0 | 24.3 | 32.7 | 37.6 | 39.3 |
| *WizardCoder* | 15B | **35.8** | **41.9** | **39.0** | **39.3** | **33.5** | **34.0** | **33.7** | **27.1** |
| *WizardCoder* | 34B | **44.9** | **55.3** | **47.2** | **47.2** | **39.8** | **41.5** | **44.3** | **46.2** |

Table 3: Performance of *WizardCoder 15B* and baseline models on DS-1000. All models are evaluated with the same set of hyper-parameters: temperature=0.2, top_p=0.5, max_length=1024. Scores are average pass@1 accuracy over 40 samples. Matplotlib (plt) task does not have the right context, so insertion and completion scores are identical.

| Format | Model | plt | np | pd | py | scp | sk | tf | All |
|---|---|---|---|---|---|---|---|---|---|
| | # of problems: | 155 | 220 | 291 | 68 | 106 | 115 | 45 | 1,000 |
| Completion | InCoder-6B | 28.3 | 4.4 | 3.1 | 4.4 | 2.8 | 2.8 | 3.8 | 7.4 |
| Completion | CodeGen-mono | 31.7 | 10.9 | 3.4 | 7.0 | 9.0 | 10.8 | 15.2 | 11.7 |
| Completion | Code-Cushman-001 | 40.7 | 21.8 | 7.9 | 12.4 | 11.3 | 18.0 | 12.2 | 18.1 |
| Completion | StarCoder | 51.7 | 29.7 | 11.4 | 21.4 | 20.2 | 29.5 | 24.5 | 26.0 |
| Completion | *WizardCoder* | **55.2** | **33.6** | **16.7** | **26.2** | **24.2** | 24.9 | **26.7** | **29.2** |
| Insertion | InCoder-6B | 28.3 | 4.6 | 2.9 | 4.4 | 2.8 | 3.1 | 7.8 | 7.5 |
| Insertion | StarCoder | 51.7 | 30.8 | 10.3 | 21.0 | 20.2 | 27.4 | 20.0 | 25.4 |
| Insertion | *WizardCoder* | **55.2** | **35.1** | **20.4** | **30.4** | **28.9** | **32.3** | **37.8** | **32.8** |

## 5 ANALYSIS

**Evolution Models and Rounds.** In Table 4, GPT-4 replaces GPT-3.5 for evolved rounds, boosting HumanEval Pass@1 scores to 73.8 (34B) and 62.2 (15B). Using OSS CodeLlama-Instruct-34B also proves effective, yielding scores of 70.1 (34B) and 55.5 (15B). Despite GPT-4's superior coding performance (88.4 vs. 73.2), the gain in evolved rounds is not proportional (73.8 vs. 73.2). Conversely, CodeLlama's weaker performance narrows when using *Code Evol-Instruct* (73.2 vs. 70.1), highlighting its crucial role. More experiments details are

Table 4: Different evolution execution models.

| Base Model | Evol Model | Pass@1 |
|---|---|---|
| StarCoder-15B | GPT-4 | 62.2 |
| StarCoder-15B | GPT-3.5 | 59.8 |
| StarCoder-15B | CodeLlama | 55.5 |
| CodeLlama-34B | GPT-4 | 73.8 |
| CodeLlama-34B | GPT-3.5 | 73.2 |
| CodeLlama-34B | CodeLlama-34B | 70.1 |

listed in Appendix E. Additionally, Figure 3 presents results for different data evolution rounds. All models are fine-tuned with 200 steps. Due to the limited size of the dev set of MBPP, we merged the training set and dev set, forming the MBPP-400 dev set. The experiments reveal that the highest pass@1 scores on both the MBPP-400 dev set and the HumanEval are achieved subsequent to three rounds of evolution.

---

[6]Given that this benchmark and its evaluation codes are not designed for the instruction fine-tuned models, we encounter significant challenges in aligning our 34B model with this framework. Moreover, the Codellama-34B base model does not support code insertion. Thus, we only include our 15B model results.

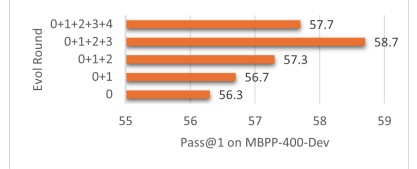

(a) Pass@1 performance on MBPP-400 dev set.

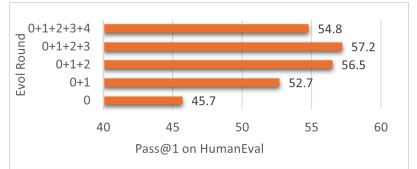

(b) Pass@1 performance on HumanEval.

Figure 3: The impact of the number of data evolution rounds.

**Complexity and Quantity.** While the enhanced performance attributed to our *Code Evol-Instruct* method has been evident in prior experiments, it remains an open question whether this performance gain is a result of an increase in the number of samples or tokens. During the evolution, each round includes more samples, and the introduction of more complex instructions inevitably leads to an increase in tokens within the training data. To address this question, we fine-tune the models using only the specific round data separately from scratch with a similar number of samples (upper part) or tokens (lower part) in Table 5.

When each round contains the same number of samples or tokens, the models trained with the seed data still lag behind the evolved rounds. Furthermore, combining data from different rounds leads to the best performance. These results suggest that the primary source of the gain is indeed attributable to our *Code Evol-Instruct* method, rather than merely an increase in samples or tokens.

**Complexity and Similarity.** Apart from the quantity analysis, we also investigate whether evolution leads to the inclusion of data more similar to the test set. To address this, we perform an analysis of the HumanEval test set. We employ test samples as queries to retrieve the top-1 sample from each evolved round's training data, utilizing the SOTA embeddings model, gte-large (Li et al., 2023b). Additionally, we employ GPT4, to provide average similarity scores between the test set and the retrieved top-1 samples. The details are shown in Appendix C.

Figure 4 illustrates that the evolution process does not yield higher similarity scores. Furthermore, similarity scores across all rounds remain relatively low. These findings indicate that the primary source of performance gain is the introduction of more complex data.

Table 5: Analysis of whether the performance gain comes from more tokens.

| Evol | #Samples | Pass@1 |
|---|---|---|
| Round 0 | 20.0k | 45.7 |
| Round 1 | 18.8k | 56.1 |
| Round 2 | 19.7k | 53.0 |
| Round 3 | 19.3k | 54.3 |
| Round 4 | 19.0k | 51.2 |

| Evol | #Tokens | Pass@1 |
|---|---|---|
| Round 0 | 2.3M | 44.5 |
| Round 1 | 2.3M | 51.8 |
| Round 2 | 2.3M | 52.4 |
| Round 3 | 2.3M | 50.0 |
| Round 4 | 2.3M | 49.4 |

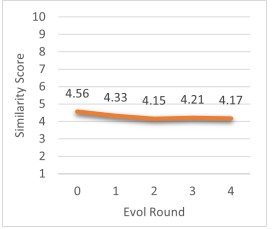

Figure 4: Average similarity scores between HumanEval samples and the top-1 retrieved data, ranging from 1 (completely different) to 10 (identical).

# 6 CONCLUSION AND FUTURE WORK

This paper introduces *WizardCoder* models, the *Code Evol-Instruct* fine-tuned Code LLMs. The experimental results demonstrate that *WizardCoder* models achieve SOTA performance surpassing all existing open-source Code LLMs on five widely recognized code generation benchmarks: HumanEval, HumanEval+, MBPP, DS-1000 and MultiPL-E. Notably, *WizardCoder 15B* model surpasses some of the well-known closed LLMs, such as Claude and Bard. Additionally, *WizardCoder 34B* achieves a HumanEval score comparable to GPT3.5 (ChatGPT) and surpasses it on the HumanEval+ benchmark. Furthermore, our analysis underscores the pivotal role of instruction complexity in enhancing performance. For future work, as depicted in Figure 2, our model still falls significantly behind the SOTA LLM, GPT4. Therefore, future work will further augment the performance of our model.

## ACKNOWLEDGMENTS

This work is partially supported by National Natural Science Foundation of China Young Scientists Fund(No. 62206233) and Hong Kong RGC ECS (No. 22200722).

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

# A  PROMPT FORMATS

In this section, we include the prompt for evaluation on different tasks.

---

**Zero-Shot Prompt for Evaluation on HumanEval and HumanEval+**

Below is an instruction that describes a task, paired with an input that provides further context. Write a response that appropriately completes the request.

### Instruction:
Create a Python script for this problem:
{Question}

### Response:

---

**Three-Shot Prompt for Evaluation on MBPP**

Below is an instruction that describes a task, paired with an input that provides further context. Write a response that appropriately completes the request.

### Instruction:
Create a Python script for this problem:
{Question}
{Test Example 1}
{Test Example 2}
{Test Example 3}

### Response:

---

**Zero-Shot Prompt for Evaluation on DS-1000 (Completion)**

Below is an instruction that describes a task, paired with an input that provides further context. Write a response that appropriately completes the request.

### Instruction:
{Question}
Complete the Python code in "...".

### Response:

---

In the case of DS-1000 (Insertion), adherence to the benchmark's specifications necessitates the utilization of StarCoder's specialized insertion symbol. Consequently, we have found it imperative to align with the same prompt format employed by StarCoder for this particular benchmark.

For the MultiPL-E benchmark, we recognized the need to align with the evaluation codes provided by bigcode-evaluation-harness.[7] Consequently, we opted to adopt the same prompt format utilized by StarCoder.

# B  BASELINES DETAILS

We include a large amount of models as our baselines. For GPT3.5 (ChatGPT)&GPT4. their results are obtained from GPT4's report and EvalPlus. The results of Code-Davinci-002, Code-Cushman-001, Codex, PaLM, PaLM 2, LaMDA, AlpahaCode, Incoder, StarCoder, LLaMa, CodeGen, CodeGeeX, CodeT5+, and InstructCodeT5+ are from StarCoder or CodeT5+'s paper. The results of Bard are evaluated with Google's API. The results of Claude are evaluated with Anthropic's API. The results

---

[7]https://github.com/bigcode-project/bigcode-evaluation-harness

of Instruct-Codegen-16B, Guanaco-65B, Falcon-40B-Instruct, and Vicuna-13B are evaluated with the open-sourced checkpoints. The results of CodeLlama-Series are from CodeLlama's paper. The results of OctoCoder are from its paper. The results of PanguCoder2 are also from its paper.

The MBPP score of StarCoder differs from that in its original paper. Through a personal contact, we were informed that StarCoder was evaluated using a cleaned and smaller version of MBPP, comprising only 397 problems, significantly fewer than the original MBPP benchmarks (500). Consequently, we conducted a re-evaluation of StarCoder using the original MBPP.

## C   SIMILARITY CHECKING AND DATA FILTERING

The prompt formats to compute the similarity score are as follow:

---

**System Prompt for Similarity Checking**

Your task is to evaluate the similarity of the two given coding tasks. Please review the two coding tasks carefully, paying close attention to the overlap in function names, code structures, topics, and contents. Once you have carefully reviewed both coding tasks, provide a similarity score between these two coding tasks. The score should range from 1 to 10 (1: completely different coding tasks; 10: identical coding tasks). You only need to provide your score. The response format is:
Score: '...'

---

**User Input for Similarity Checking**

# Task1
{task1}

# Task2
{task2}

---

To thoroughly prevent data leakage from the test datasets to the training dataset, we implemented an additional data filtering step. Utilizing the SOTA embeddings model, gte-large, we treated all test samples as queries to extract the top 5 samples from the training data. Following this, GPT-4 was employed to evaluate the similarity between the retrieved samples and the test sample. The task for GPT-4 is simplified to a binary decision—either a "yes" or "no" indicating a match. In case of a positive match, the sample is excluded from the training data.

## D   EVOL EXAMPLES

In this section, we present some evolved examples to elucidate the influence exerted by our *Code Evol-Instruct*.

Example 1:

- Round 0: Write a MongoDB query to select all documents in a collection where the field 'category' is 'clothes'.
- Round 1: Write a MongoDB query to select all documents in a collection where the field 'category' is 'clothes' and the 'brand' field is not equal to 'Nike'.
- Round 2: Write a MongoDB query to select all documents in a collection where the field 'category' is 'clothes' and the 'brand' field is not equal to 'Nike', and the 'price' field is greater than or equal to 100 and less than or equal to 500.
- Round 3: Write a MongoDB query to select all documents in a collection where the field 'category' is 'clothes' and the 'brand' field is not equal to 'Nike', and the 'price' field is greater than or equal to 100 and less than or equal to 500, and the 'color' field is either 'red' or 'blue'. Additionally, sort the documents in descending order by the 'date_added' field and limit the result to the first 10 documents.

Example 2:

- Round 0: Using Tensorflow, set up a model to predict the classification of a handwritten digit.
- Round 1: Using Tensorflow, set up a model to predict the classification of a handwritten digit with the constraint that the model must be trained using only 1000 samples from the MNIST dataset.
- Round 2: Using TensorFlow, set up a model to predict the classification of a handwritten digit with the constraint that the model must be trained using only 500 samples from the MNIST dataset. Additionally, implement transfer learning to incorporate the features learned from a pre-trained model.
- Round 3: Using TensorFlow and the MNIST dataset, develop a convolutional neural network (CNN) to accurately classify handwritten digits. However, in this new version of the task, you must limit your training to a maximum of 500 images. Furthermore, incorporate transfer learning by using a pre-trained model to extract features from the data and improve your model's accuracy.

Example 3:

- Round 0: Create a class in Java that can store a list of students and the courses they are enrolled in.
- Round 1: Create a Java program that manages a student database with courses, including options to add, remove, and search for students, as well as courses for each student. The program should also be able to display statistics on the number of students and courses.
- Round 2: Develop a Java program that manages a student database containing courses and grades. Your program should allow users to add, remove, and search for students, as well as add or drop courses for each student. Furthermore, it should display statistics on the number of students and courses, and calculate the average grade for each student. Ensure that your program implements appropriate error-handling mechanisms.
- Round 3: Develop a Java program to manage a library system where users can check-in, check-out and search for books in the library. The system should allow users to add new books and delete existing ones from the library. The program should also provide statistics on the number of books checked-in and checked-out, and calculate the average reading time for each book. Ensure that your program implements appropriate error-handling mechanisms and enforces data integrity constraints such as not allowing users to check-out books that have already been checked-out.

# E  USING DIFFERENT EVOLUTION EXECUTION MODELS

We explored using the open-source models (OSS) CodeLlama-Instruct-34B for generating evolved instructions. However, it demonstrated relatively low coding performance in response generation. To address this, we fine-tuned it using our code-alpaca dataset and utilized this model for response generation.

# F  INTERACTION EXAMPLES

Table 6 7 8 showcases examples of interactions with our *WizardCoder*. The examples demonstrate that our model consistently generates accurate responses accompanied by clear explanations.

Table 6: Examples of interaction with our *WizardCoder*.

| Instruction | Response |
|---|---|
| Write a Python function to tell me what the date is today. | Here's a Python function that tells you what the date is today:

```python
import datetime

def get_date():
    now = datetime.datetime.now()
    return now.strftime("%A, %B %d, %Y")

print(get_date())
```

This function uses the datetime module to get the current date and time. It then uses the strftime() method to format the date as a string in the format "Day, Month Date, Year". Finally, it returns the formatted string.

You can call this function from your Python code to get the current date and time. |
| I do not know how to integrate a Python function numerically. Please write a code for me. | Sure, here's an example code for integrating a Python function numerically using the trapezoidal rule:

```python
def integrate(f, a, b, n):
    h = (b - a) / n
    s = (f(a) + f(b)) / 2
    for i in range(1, n):
        s += f(a + i * h)
    return h * s
```

This code takes in a function f, the lower bound a, the upper bound b, and the number of subintervals n. It then calculates the width of each subinterval h, and uses the trapezoidal rule to approximate the integral of f between a and b. The result is returned as a float.
To use this function, you would call it with the function you want to integrate, the lower bound, the upper bound, and the number of subintervals you want to use. For example:

```python
def f(x):
    return x**2

result = integrate(f, 0, 1, 1000)
print(result)
```

This would integrate the function f(x) = x**2 between 0 and 1 using 1000 subintervals, and print the result. |

Table 7: Examples of interaction with our *WizardCoder*.

| Instruction | Response |
|---|---|
| List the prime numbers between 20 and 30 with Java. | Here's the Java code to list the prime numbers between 20 and 30: |

```java
public class PrimeNumbers {
    public static void main(String[] args) {
        for (int i = 20; i <= 30; i++) {
            boolean isPrime = true;
            for (int j = 2; j < i; j++) {
                if (i % j == 0) {
                    isPrime = false;
                    break;
                }
            }
            if (isPrime) {
                System.out.print(i + " ");
            }
        }
    }
}
```

Output:

```
23 29
```

Explanation:
- We use a nested for loop to iterate through all the numbers between 20 and 30.
- For each number, we use another for loop to check if it is prime or not.
- If a number is divisible by any number other than 1 and itself, then it is not prime.
- If a number is not divisible by any number other than 1 and itself, then it is prime.
- We print the prime numbers to the console.

Table 8: Examples of interaction with our *WizardCoder*.

| Instruction | Response |
|---|---|
| Write a simple neural network with pytorch | Here is a simple neural network with pytorch: |

```python
import torch
import torch.nn as nn

class Net(nn.Module):
    def __init__(self):
        super(Net, self).__init__()
        self.fc1 = nn.Linear(2, 4)
        self.fc2 = nn.Linear(4, 1)
        self.sigmoid = nn.Sigmoid()

    def forward(self, x):
        x = self.fc1(x)
        x = self.sigmoid(x)
        x = self.fc2(x)
        return x

net = Net()
criterion = nn.MSELoss()
optimizer = torch.optim.SGD(net.parameters(), lr=0.01)

# Training
for epoch in range(1000):
    inputs = torch.randn(10, 2)
    targets = torch.randn(10, 1)
    optimizer.zero_grad()
    outputs = net(inputs)
    loss = criterion(outputs, targets)
    loss.backward()
    optimizer.step()

# Testing
inputs = torch.randn(10, 2)
outputs = net(inputs)
print(outputs)
```

In this example, we define a neural network with two fully connected layers and a sigmoid activation function. We use the mean squared error loss function and the stochastic gradient descent optimizer. We train the network for 1000 epochs and then test it with some randomly generated inputs. The output of the network is a tensor of predicted values.

