# OpenReview forum: "WizardCoder: Empowering Code Large Language Models with Evol-Instruct"
_ICLR.cc/2024/Conference — ICLR 2024 poster_

### Official Review · Reviewer_FnZW · 2023-11-01

**Soundness:** 4 excellent
**Presentation:** 4 excellent
**Contribution:** 3 good
**Rating:** 8
**Confidence:** 4

**Summary:**

This paper presents a novel approach called "Code Evol-Instruct", which adapts the "Evol-Instruct" method to the domain of code, enhancing the performance of open-source Code LLMs.The authors use Code Evol-Instruct to evolve basic code instruction data and then fine-tune existing open-source Code LLMs, resulting in the creation of "WizardCoder" models to improve the ability of these models to generate high-quality code based on more complex instructions. The WizardCoder outperforms other open-source models and even rivals larger closed-source LLMs in certain coding tasks.

**Strengths:**

1. Introduction of Code Evol-Instruct, an innovative method for enhancing open-source Code LLMs, which significantly improves their performance in code generation tasks.
2. Even the smaller 15B version of WizardCoder outperforms larger closed-source LLMs like Claude and Bard on certain benchmarks, while the 34B version achieves performance comparable to GPT-3.5 (ChatGPT) and surpasses it on specific benchmarks.

**Weaknesses:**

1. In section 4.3, Table 1, Result of pass@1(%) on HumanEval and MBPP, the result of pass@1(%) on MBPP is not as good as the result on HumanEval. HumanEval only tests a narrow distribution and can be overfitted. The benchmark needs to be upgraded.

**Questions:**

1. In section 3.2, TRAINING WizardCoder," We iteratively employ the Code Evol-Instruct technique on this dataset consisting of around 20k samples to produce evolved data."  What is the number of iterations and the criteria for completion?
2. In section 3, when iterating the Code Eval-Instruct on the sample dataset, what model is used to generate evolved data? Does the difference in the quality of pre-trained model outputs lead to different results?

---

> ### Author Response · Authors · 2023-11-20
> **Response to Reviewer FnZW**
>
> Thank you for your thorough review! We are grateful for your professional feedback. We are particularly encouraged that you think our method is innovative and our paper is good!
>
> > **Weakness**: In section 4.3, Table 1, Result of pass@1(%) on HumanEval and MBPP, the result of pass@1(%) on MBPP is not as good as the result on HumanEval. HumanEval only tests a narrow distribution and can be overfitted. The benchmark needs to be upgraded.
>
> We acknowledge the need for a more comprehensive set of benchmarks in the coding domain. Our evaluation extends beyond HumanEval, encompassing a diverse set of benchmarks, including MBPP, DS-1000, and MultiPl-E. Notably, when compared to SOTA open-source software (OSS) models, our models showcase substantial improvements across various benchmarks. Specifically, our models achieve approximately a 20-point improvement on HumanEval, a 5.2-point improvement on MBPP, a 7.4-point improvement on DS-1000 (insertion), and an average 9.3-point improvement on MultiPL-E. These significant enhancements underscore the effectiveness of our work. Additionally, it's worth noting that Codellama-instruct only achieves a 2-point improvement compared to Codellama on MBPP.
>
> > **Question 1**: In section 3.2, TRAINING WizardCoder," We iteratively employ the Code Evol-Instruct technique on this dataset consisting of around 20k samples to produce evolved data." What is the number of iterations and the criteria for completion?
>
> We conducted a total of 4 iterations. Figure 3a reveals that the optimal performance is achieved after 3 rounds of evolutions on the dev set. Based on this observation, we made the decision to discontinue further iterations.
>
> > **Question 2**: In section 3, when iterating the Code Eval-Instruct on the sample dataset, what model is used to generate evolved data? Does the difference in the quality of pre-trained model outputs lead to different results?
>
> Please refer to the section "2. Common Response to using different evolution execution models" in the Overall Response. We have provided the results of using GPT-4 as the evolution execution model.
>
> We have mentioned in Section 4.2 that we use GPT3.5 (gpt3.5-turbo) to generate evolved data. We conduct additional experiments, replacing GPT3.5 (gpt3.5-turbo) with GPT-4 to generate evolved rounds. HumanEval Pass@1 scores increased from 73.2 to 73.8 for 34B and from 59.8 to 62.2 for 15B. Using a superior LLM does contribute to better evolved data; however, the improvement is not strongly correlated with the coding performance of the evolution executing model (88.4 for GPT4 and 73.2 for GPT3.5).
>
> In summary, we found your comments insightful and hope that our responses have addressed any questions. If you have additional questions, please feel free to reach out. Once again, we sincerely appreciate your thoughtful review!

---

> > ### Comment · Reviewer_FnZW · 2023-11-23
> >
> > Thanks for the clarification.
> >
> > I have read the response. I know that the novelty of this paper is quite limited as it is an adaptation of evol-instruct for code. Personally, I think studying how to use current instruction tuning techniques to improve the performance of code foundation models is important because the coding ability of foundation models has a broad impact in the field of programming, embodied AI, and LLM agents (like tool-using and tool generation). This paper at least makes a successful attempt on evol-instruct for code. So I keep my score for acceptance.

---

> > > ### Author Response · Authors · 2023-11-23
> > > **Thank You**
> > >
> > > Thank you for your thoughtful consideration and understanding of our paper. We appreciate your recognition of the importance of exploring instruction tuning techniques for enhancing the performance of code foundation models. Your support means a lot to us.

---

### Official Review · Reviewer_QRie · 2023-11-02

**Soundness:** 3 good
**Presentation:** 3 good
**Contribution:** 2 fair
**Rating:** 5
**Confidence:** 4

**Summary:**

The paper extends the idea of instruction fine-tuning to code LLMs. The proposed approach uses GPT-3.5 to evolve existing instruction data to generate complex and diverse set of examples in an iterative manner. Using the synthetic evolving data followed by instructions, the authors fine-tune SOTA coda LLMs. The results show improved performance of the models for a wide range of tasks.

**Strengths:**

$\mathtt{+}$ I think overall exploring ideas around how we can improve the efficacy of LLMs for different application is interesting.

$\mathtt{+}$ Improving SOTA using the proposed instruction-tuning method is valuable and opens up new direction. The ablation studies further help to expand how and to what extend each technique helps (with some caveats that I will expand in the question section).

**Weaknesses:**

$\mathtt{-}$ It is not clear how the authors came up with the list of heuristics for data evolution. This unclarity makes such approaches less applicable to wide range of tasks.

$\mathtt{-}$ While the ablation studies in the main body provides some insights on the efficacy of the technique (additional clarification in the questions/recommendation section).

$\mathtt{-}$ While the idea is interesting, but it seems very incremental compared to prior work and the contributions are limited.

**Questions:**

(Q1) Table 4, the Pass@1 score decreases as the number of iterations increases. Do you have any insights as why this is happening?

(Q2) How the performance of your model increases as you use less capable models for generating evolving data? In addition, do you think we can use the same model to generate the data for next iteration?

(Q3) Can you clarify how did you come up with the heuristics? Did you eliminate any of them based on the final outcome? Was the process a trial-error approach or there were some insights behind making any of the decisions?

---

> ### Author Response · Authors · 2023-11-20
> **Response to Reviewer QRie**
>
> Thank you for your thorough review! We are grateful for your dedication throughout the review period. Your positive comments, "proposed instruction-tuning method is valuable" and "the idea is interesting", are sincerely appreciated.
>
> > **Weakness 1**: It is not clear how the authors came up with the list of heuristics for data evolution. This unclarity makes such approaches less applicable to wide range of tasks.
>
> > **Question 3**: Can you clarify how did you come up with the heuristics? Did you eliminate any of them based on the final outcome? Was the process a trial-error approach or there were some insights behind making any of the decisions?
>
> We created these five heuristic methods to increase the complexity of code instructions, aligning with the characteristics of the code domain and previous research on code models' limitations. The first three heuristics naturally escalate coding task difficulty by incorporating additional requirements, replacing common requirements, or introducing extra reasoning steps. This aligns with observed trends on platforms like LeetCode, where tasks with more requirements or intricate reasoning are considered more challenging. The fourth heuristic introduces erroneous code as an adversarial sample, inspired by prior research in attacking pre-trained code models [1][2], serving to assess the robustness of the code LLM. The fifth heuristic, emphasizing time and space complexity, aligns with the significance of optimization in the coding domain. Previous studies [3] indicate that tasks requiring optimized program performance pose heightened difficulty, making this direction a valuable avenue for complexity improvement.
>
> [1] Yang, Zhou et al. “Natural Attack for Pre-trained Models of Code.” 2022 IEEE/ACM 44th International Conference on Software Engineering (ICSE).
>
> [2] Jha, Akshita et al. “CodeAttack: Code-based Adversarial Attacks for Pre-Trained Programming Language Models.” AAAI 2022.
>
> [3] Madaan, A. et al. "Learning Performance-Improving Code Edits." (arxiv, 2023)
>
> > **Weakness 3**: While the idea is interesting, but it seems very incremental compared to prior work and the contributions are limited.
>
> Please refer to "1. Common Response to novelty and contributions" in the Overall Response regarding the novelty and contributions of our work.
>
> > **Question 1**: Table 4, the Pass@1 score decreases as the number of iterations increases. Do you have any insights as why this is happening?
>
> In the upper part of Table 4, the Pass@1 scores exhibit fluctuations rather than a monotonic decrease; it's important to highlight that only the fourth round notably deteriorates. This decline may be attributed to two key factors. First, continual evolution could lead to overly complicated instructions, rendering them unreasonable. Second, beyond a certain round, instructions may become excessively complex, posing a significant challenge to the evolution executing model (i.e., gpt3.5-turbo) in generating suitable responses.
>
> In the lower part of Table 4, **the decreasing scores are normal**. As instructions become more complex, the length of samples increases, leading to a reduction in the number of samples in each round to control for the same token count.
>
> It's crucial to recognize that individual metrics in Table 4 for a specific round with complex data might not fully represent the overall impact on the final model performance. Figure 3 provides insights, indicating that merging different complex-level data can result in performance gains.
>
> > **Question 2**: How the performance of your model increases as you use less capable models for generating evolving data? In addition, do you think we can use the same model to generate the data for next iteration?
>
> Please refer to "2. Common Response to using different evolution execution models" in Overall Response. We have provided the results of using less capable models as evolution execution models.
>
> We explored using the open-source models (OSS) CodeLlama-Instruct-34B for generating evolved instructions. However, it demonstrated relatively low coding performance in response generation. To address this, we fine-tuned it using the code-alpaca dataset and utilized this model for response generation. The OSS-based Code Evol-Instruct tuned models also exhibited performance improvements compared to the code-alpaca fine-tuned model, increasing from 65.9 to 70.1 for 34B and from 45.7 to 55.5 for 15B. Despite CodeLlama exhibiting weaker performance than GPT3.5 (gpt3.5-turbo), the performance gap significantly diminishes when utilizing our Code-Evol Instruct to generate evolved rounds (73.2 vs. 70.1).
>
> Generating data for the next iteration using the same model is challenging because the OSS code model is trained to generate code responses and not to generate evolved instructions.
>
> Overall, we hope that our responses clarify your questions. Please let us know if you have further questions, and thank you again for your review!

---

### Official Review · Reviewer_fiHu · 2023-11-04

**Soundness:** 2 fair
**Presentation:** 2 fair
**Contribution:** 3 good
**Rating:** 6
**Confidence:** 5

**Summary:**

The paper proposed a new way, Code Evol-Instruct, to fine-tune an LLM for coding tasks. The method starts from a set of existing code questions, then uses another LLM (GPT 3.5, in this case) to add one more step of complexity or difference to make a new question. Detailed prompts for evol-instruct are provided in the paper. 4 rounds of evol-instruct expanded the original 20k Code Alpaca instruction set to 87k. The WizardCoder model fine-tuned with this set ranked only behind GPT-4 in HumanEval and other popular coding benchmarks, beating other major open and closed-sourced models.

The paper also analyzes the effectiveness of the evol-instruct set from different rounds of expansion. The results show that generally more data leads to better performance (except with the addition of round 4), the same amount of tokens from more complex instructions works better, and the data expansion didn't create more similar data to the test set.

**Strengths:**

- The method works. It produced a top-performing open source code model that surpasses bigger and closed-source models in multiple open evaluations. This is the biggest strength of the paper and the value to the research community.

- Paper provided extensive comparison with existing models and between different rounds of evol-instruct expansions.

**Weaknesses:**

- Missing some key details. Where does the new coding solution come after expanding the instruction? Did we use the base model itself, or GPT-3 to generate them? Did we do any deduplication of the expanded instructions? Did we verify the quality of new instructions by executing the code generated?

Update: based on the author's feedback, the coding solutions in the training data are from GPT-3.5-turbo. This presents a risk of data leakage, that WizardCoder becomes an implicitly distilled model of GPT-3.5-turbo on a specific capability.

- Weak analysis. Why does round 4 have a negative impact on the quality? The paper mentioned "some evolution process will fail" but provided no detail. Other analysis related questions will be in the "questions" section.

- Writing quality can be improved. The paper uses more sentences to repeatedly claim WizardCoder's performance (which is indeed impressive) but vague on implementation details and analysis.

- Inconsistency in the data. Figure 2 reported GPT-4 has a HumanEval pass@1 of 88.4, while Table 1 showed 67.0.

**Questions:**

- Where does the new coding solution come after expanding the instruction?  Did we use the base model itself, or GPT-3 to generate them?

- Did we do any deduplication of the expanded instructions?

- Did we verify the quality of new instructions by executing the code generated?

- Figure 1 lower plot. It's better to use pattern instead of color to separate two data classes to make it more friendly for color-blindness.

- Section 1 paragraph 2. Reviewer thinks "Code" need not to have capitalized first letter.

- Section 2 Related Work. We should also mention CodeLlama's self-instruct work and compare the differences. https://arxiv.org/abs/2308.12950

- Page 4, Code Evolution Heuristic Methods table. Please explain the 4th instruct's purpose and examples of expansion. "Provide a piece of erroneous code as a reference to increase misdirection". It's not clear to the Reviewer what this is doing and why it would be effective.

- Page 4, Code Evolution Heuristic Methods table. Last prompt about increasing time/space complexity: how can we verify the response actually achieved it?

- Section 5 paragraph 1, "amalgamate the training set", what about using a simpler word like  "merge the training set"?

- Section 5 paragraph 2. The Reviewer assumes more rounds equals more complexity. But it's better to make it explicit and provide some examples to show what does complexity actually mean in this context..

- Section 5 "Complexity and Similarity". This paragraph didn't really talk about complexity, unless more rounds = more complexity?

- Table 4. It's better to show the confidence interval on this table, if possible, since 400 examples is not that large a test set.

- Section 6: conclusion. ".. the pivotal role of instruction complexity in enhancing coding performance". Reviewer doesn't think this claim holds well given the analysis in the paper. For example, in Table 4, later rounds (assuming more complexity) lead to a lower performance (or statistically insignificant changes).

---

> ### Author Response · Authors · 2023-11-20
> **Response to Reviewer fiHu [1/3]**
>
> Thank you for your insightful reviews! Your professional feedback provides valuable guidance for writing a more comprehensive and competitive paper. We are particularly encouraged that you found our WizardCoder's performance impressive!
>
> > **Weakness 1**: Missing some key details. Where does the new coding solution come after expanding the instruction? Did we use the base model itself, or GPT-3 to generate them? Did we do any deduplication of the expanded instructions? Did we verify the quality of new instructions by executing the code generated?
>
> The coding solution is generated by the same model that produces the evolved instructions, specifically gpt3.5-turbo. The seed data, employing the self-instruct technique, is inherently deduplicated. Given that the expanded instructions derive from distinct coding tasks, there is no need for deduplication in the subsequent evolved process. We have tried to execute all the generated Python codes. All of them can run without any compile errors.
>
> > **Weakness 2**: Weak analysis. Why does round 4 have a negative impact on the quality? The paper mentioned "some evolution process will fail" but provided no detail. Other analysis related questions will be in the "questions" section.
>
> Our empirical results reveal a potential saturation point from Code Evol-Instruct after a specific number of rounds, namely 4. This decline may be attributed to two key factors. First, continual evolution could lead to overly complicated instructions, rendering them unreasonable. Second, beyond a certain round, instructions may become excessively complex, posing a huge challenge to the evolution executing model (i.e., gpt3.5-turbo) to generate suitable responses.
>
> > **Weakness 3**: Writing quality can be improved. The paper uses more sentences to repeatedly claim WizardCoder's performance (which is indeed impressive) but vague on implementation details and analysis.
>
> We present the implementation details in Section 4.2, with the evolution prompts discussed in Section 3.1 and the training process outlined in Section 3.2. The prompt formats are provided in Appendix A, baseline details in Appendix B, and data filtering details in Appendix C.
>
> Furthermore, our analysis in Section 5 encompasses various aspects, such as the impact of different numbers of evolution rounds, the relationships between complexity and quantity, and the correlations between complexity and similarity.
>
> > **Weakness 4**: Inconsistency in the data. Figure 2 reported GPT-4 has a HumanEval pass@1 of 88.4, while Table 1 showed 67.0.
>
> In Footnote 5, Page 6, we have discussed why the scores in Figure 2 and Table 1 are different. In Figure 2, the scores are reported by the Eval-Plus leaderboard with OpenAI's APIs. In Table 1, the scores are reported by OpenAI's report and CodeLlama's paper. **In order to make the conclusion more rigorous, all the claims regarding GPT3.5(ChatGPT) and GPT4 in our paper rely on results obtained through OpenAI's APIs.**

---

> ### Author Response · Authors · 2023-11-20
> **Response to Reviewer fiHu [2/3]**
>
> > **Question 1**: Where does the new coding solution come after expanding the instruction? Did we use the base model itself, or GPT-3 to generate them?
>
> We use the same model (gpt3.5-turbo) to generate the code solutions.
>
> > **Question 2**: Did we do any deduplication of the expanded instructions?
>
> The seed data, employing the self-instruct technique, is inherently deduplicated. Given that the expanded instructions derive from distinct coding tasks, there is no need for deduplication in the subsequent evolved process.
>
> > **Question 3**: Did we verify the quality of new instructions by executing the code generated?
>
> We have tried to execute all the generated Python codes. All of them can run without any compile errors.
>
> > **Question 4**: Figure 1 lower plot. It's better to use pattern instead of color to separate two data classes to make it more friendly for color-blindness.
>
> Thank you for your suggestion. We updated Figure 1 lower plot with different patterns.
>
> > **Question 5**: Section 1 paragraph 2. Reviewer thinks "Code" need not to have capitalized first letter.
>
> Thank you for your suggestion. We updated Section 1 paragraph 2.
>
> > **Question 6**: Section 2 Related Work. We should also mention CodeLlama's self-instruct work and compare the differences. https://arxiv.org/abs/2308.12950
>
> We have mentioned the CodeLlama-Series models in the related work. The CodeLlama-Instruct model is also one of the baseline models in our experiments. We include extra discussion of this model in the related works section.
>
> > **Question 7**: Page 4, Code Evolution Heuristic Methods table. Please explain the 4th instruct's purpose and examples of expansion. "Provide a piece of erroneous code as a reference to increase misdirection". It's not clear to the Reviewer what this is doing and why it would be effective.
>
> For a specific instruction, this method includes an error code in it. For example, the original instruction is: `Write a Python function to tell me what the date is today.`
>
> The evolved instruction is:
> `Write a Python function to tell me what the date is today, using the datetime module. The function should return a string in the format of “Today is YYYY-MM-DD”. For example, if today is November 17, 2023, the function should return “Today is 2023-11-17”. As a reference, here is a piece of erroneous code that does not work as intended:`
> ```python
> import datetime
> def get_date():
>     today = datetime.date.today()
>     return "Today is " + today
> ```
>
> This erroneous code serves as an adversarial sample designed to challenge the code LLM, akin to previous works focused on attacking pre-trained code models. [1][2]
>
> [1] Yang, Z., Shi, J., He, J., & Lo, D. (2022). Natural Attack for Pre-trained Models of Code. 2022 IEEE/ACM 44th International Conference on Software Engineering (ICSE), 1482-1493.
>
> [2] Jha, A., & Reddy, C.K. (2022). CodeAttack: Code-based Adversarial Attacks for Pre-Trained Programming Language Models. AAAI Conference on Artificial Intelligence.
>
> > **Question 8**: Page 4, Code Evolution Heuristic Methods table. Last prompt about increasing time/space complexity: how can we verify the response actually achieved it?
>
> This heuristic, which underscores time and space complexity, aligns with the importance of optimization in the coding domain. Previous studies [3] suggest that tasks demanding optimized program performance present heightened difficulty, rendering this direction a valuable avenue for complexity improvement. Ideally, all responses should achieve 100% compliance with the specified complexity requirements. Given our use of GPT3.5(gpt3.5-turbo) to generate responses, we can express a reasonable level of confidence that the responses are well-aligned with the stipulated requirements.
>
> [3] Madaan, A., Shypula, A., Alon, U., Hashemi, M., Ranganathan, P., Yang, Y., Neubig, G., & Yazdanbakhsh, A. (2023). Learning Performance-Improving Code Edits. ArXiv, abs/2302.07867.

---

> ### Author Response · Authors · 2023-11-20
> **Response to Reviewer fiHu [3/3]**
>
> > **Question 9**: Section 5 paragraph 1, "amalgamate the training set", what about using a simpler word like "merge the training set"?
>
> Thank you. We updated it in Section 5 paragraph 1.
>
> > **Question 10**: Section 5 paragraph 2. The Reviewer assumes more rounds equals more complexity. But it's better to make it explicit and provide some examples to show what does complexity actually mean in this context.
>
> Yes. More rounds equals more complexity. We have provided some evol examples in Appendix D.
>
> > **Question 11**: Section 5 "Complexity and Similarity". This paragraph didn't really talk about complexity, unless more rounds = more complexity?
>
> Yes.
>
> > **Question 12**: Table 4. It's better to show the confidence interval on this table, if possible, since 400 examples is not that large a test set.
>
> The primary objective of Table 4 is to underscore the pivotal role of instruction complexity in enhancing coding performance. The table effectively demonstrates that models trained with any round of round 1-4 data (with evolved complexity) outperform round 0 data (without evolved complexity) with a substantial gap. Given the clear difference in performance between any evolved round and non-evolved data, the presentation of confidence intervals is unnecessary to prove that evolved data is better.
>
> > **Question 13**: Section 6: conclusion. ".. the pivotal role of instruction complexity in enhancing coding performance". Reviewer doesn't think this claim holds well given the analysis in the paper. For example, in Table 4, later rounds (assuming more complexity) lead to a lower performance (or statistically insignificant changes).
>
> We appreciate your constructive comments. What we intended to convey is that making instruction more complex benefit code model fine-tuning. To better substantiate our claim and eliminate possible misunderstandings, we would like to further explain Table 4 and the corresponding experimental results. Table 4 demonstrates that models trained with any round of round 1-4 data (incorporating evolved complexity) exhibit significantly better performance than round 0 data (devoid of evolved complexity). Though rounds 1-4 are not a monotonous process with increasing complexity, they are all better than the code-alpaca data with low complexity. This indicates that complexity helps to achieve good performance. Figure 3 further supports this by showcasing that the combination of rounds 0-3 achieves the best performance, indicating the beneficial impact of introducing instruction complexity. We sincerely hope that the clarification could address your concern.

---

> > ### Comment · Reviewer_fiHu · 2023-11-22
> > **Data leakage from GPT-3 is the main concern.**
> >
> > Thanks for the detailed answers!
> >
> > I noticed that the instructions and particularly the responses (code) are from GPT-3.5 This could mean an implicit distillation from GPT-3.5 which is considered a strong coding model. We should consider redo the same experiment with the same family of models (or the base model of wizardcoder itself) to make sure this is not some implicit data leakage from GPT-3.5

---

> > > ### Author Response · Authors · 2023-11-22
> > > **Response to the Comment of Reviewer fiHu**
> > >
> > > Thank you for your review. If any of your concerns haven't been fully addressed or if you still have concerns about the GPT 3.5 leakage issue, we welcome further discussion at any time.

---

> > > > ### Comment · Reviewer_fiHu · 2023-11-22
> > > >
> > > > Thank you for the additional explanation about experiment without GPT 3.5 and it's encouraging to see the improvements without explicitly using GPT-3.5.
> > > >
> > > > I will adjust the rating but please make sure the non-GPT-3.5 version of the result is clearly presented in the final manuscript, as this is my main concern. Thanks for the good work!

---

> > > > > ### Author Response · Authors · 2023-11-23
> > > > > **Thank You**
> > > > >
> > > > > Thank you for reviewing our paper and adjusting the rating. We've included the non-GPT-3.5 results in Table 4 of Section 5, with detailed discussions in "Section 5 Evolution Models and Rounds". Your professional comments serves as invaluable guidance, contributing to the enhancement of our paper. We genuinely appreciate the time and thoughtful consideration you have dedicated to our work.

---

> ### Author Response · Authors · 2023-11-22
> **Response to the Comment of Reviewer fiHu**
>
> Thank you for taking the time to review our rebuttals, and we appreciate your ongoing consideration of our submission.
>
> We have conducted additional experiments to replace the evolution execution models with the same family of our models. The results are presented in the "Overall Response: 2. Common Response to using different evolution execution models". For your convenience, we also include some results here:
>
> |  Base Model  | Evol Model | HumanEval    |
> | ----- |----| ---- |
> | CodeLlama-34B | GPT3.5(ChatGPT) | 73.2 |
> | CodeLlama-34B | CodeLlama | 70.1 |
>
> As you suggested, we explored using the same family of open-sourced models (OSS), specifically the CodeLlama-Instruct-34B, for executing the entire process of Code Evol-Instruct. The OSS-base Code Evol-Instruct tuned models also exhibited performance improvements compared to the code-alpaca fine-tuned model, increasing from 65.9 to 70.1. **This demonstrates that our proposed method is still effective without GPT-3.5.**
>
> To eliminate any possible misunderstandings, we are willing to explain our experimental setup for you in detail. Our experimental setup is reasonable lies in that our baseline code-alpaca is also generated from the GPT3.5 model but using a different method (i.e., self-instruct). As shown in the upper part of Table 4 of Section 5 in the main submission, when ensuring a similar amount of training data, the model (rounds 1-4) fine-tuned from Code Evol-Instruct created data is superior to the model (rounds 0) fine-tuned with code-alpaca data.
>
> Overall, our Code Evol-Instruct emerges as the pivotal factor in driving performance enhancement, rather than some implicit data leakage from GPT-3.5. We sincerely hope that the clarification could address your concern.

---

### Official Review · Reviewer_6CYQ · 2023-11-09

**Soundness:** 3 good
**Presentation:** 2 fair
**Contribution:** 3 good
**Rating:** 6
**Confidence:** 4

**Summary:**

WizardCoder proposes an instruction fine-tuning method for code - Code Evol-Instruct where a large synthetic dataset is created using a seed set of code tasks (Code Alpaca) which are evolved by GPT-3.5 to increase their difficulty and complexity in multiple rounds. When fine-tuned using this dataset, publicly available models like CodeLlama and StarCoder are shown to outperform all other open sourced alternatives, and are comparable to some of the closed source ones on many different benchmarks on code generation like HumanEval, MBPP and DS-1000.

**Strengths:**

- Clever instruction finetuning idea on creating datasets synthetically using gpt-3.5 and a small set of seed tasks
- Exhaustively tested on different programming languages, impressive performance gains using publicly available models (StarCoder and CodeLlama-34B) across several benchmarks. -HumanEval+, MBPP, MultPL-E, and DS-1000

**Weaknesses:**

- While results on code benchmarks are impressive, novelty of the scientific methodology itself is quite limited as it is an adaptation of Evol-Instruct for Code.
- Missing human assessment - It is not clear how useful the final fine-tuned model is outside the benchmarks that focus exclusively on functional correctness. Model hasn't been tested on developer productivity tasks like completion, code refinement.
- Not clear if data leakage has been prevented. Does the evolved data or seed data overlap with HumanEval or other benchmarks' test set?
- The paper assumes the reader to be familiar with Evol Instruct, and does not provide sufficient context to follow the method.

**Questions:**

- Figure 3: Why are results from evol round 0 + 1 + 2 + 3 + 4 worse than 0 + 1 + 2 + 3? Do returns from EvolInstruct start to diminish or turn negative after a certain number of rounds? This is not discussed or explored in this paper.

- Analysis - Complexity and Quantity: This section is not clear to me. What are the results shown in Table 4? How does Table 4 make one conclude that the gains are not due to increase in samples or tokens? Are the 4 rows in Table 4 pass@1 metric for model checkpoints when sequentially trained on each round?

- How dependent is the performance of Evol-Instruct on the evolving model? Will the performance substantially improve if we use GPT-4 to generate evolved rounds? Will it worsen if the evolving model is changed to other open sourced alternatives?

- Section 3.1: We optimised the evolutionary instructions by eliminating deepening and complicating inputs, as well as In-Breadth Evolution. What does this mean? Not clear how 1 and 2 are different, can you describe precisely with an example?

- Can you provide details of the compute infrastructure involved and hyperparameters in fine-tuning? (number of GPU hours, batch-size, sequence length) and other details)

- The abstract (and some places in the paper) comment on model sizes of Claude and Bard as 'largest closed-source LLMs', this is not verifiable, so I suggest the authors re-word this without commenting on their model size and avoid other such colloquial writing present in the paper to improve its soundness.

Typos:
- Intro Para 3: several key adaptions --> adaptations

---

> ### Author Response · Authors · 2023-11-20
> **Response to Reviewer 6CYQ [1/3]**
>
> Thank you for your insightful reviews! Your professional feedback provides valuable guidance for writing a more comprehensive and competitive paper. We are particularly encouraged that you found our results on code benchmarks impressive!
>
>
> > **Weakness 1**: While results on code benchmarks are impressive, novelty of the scientific methodology itself is quite limited as it is an adaptation of Evol-Instruct for Code.
>
> Please refer to the common response 1 regarding the novelty and contributions of our work.
>
>
> > **Weakness 2**: Missing human assessment - It is not clear how useful the final fine-tuned model is outside the benchmarks that focus exclusively on functional correctness. Model hasn't been tested on developer productivity tasks like completion, code refinement.
>
> Tasks related to developer productivity can serve as useful assessments of the practical applicability of code models in real-world industrial scenarios. However, even the most advanced LLM, GPT-4, faces challenges when applied to real-life industrial coding tasks. Thus, the most popular evaluation method for code LLMs research is the execution-based function-level code snippet generation, where the generated code snippet/patch undergoes unit tests to ensure functional correctness. We follow this research line to explore methods to improve the performance of function-level code snippet generation. **Our evaluation strategy aligns with the previous and contemporary SOTA works in this research line, such as Starcoder and Codellama, as well as works accepted by ICLR23 and KDD23, including CodeGen and CodeGeeX**. These works similarly exclude human assessment of code.
>
> The chosen evaluation tasks, namely HumanEval, MBPP, DS-1000, and MultiPL-E, collectively cover a spectrum of diverse code completion and insertion scenarios. Specifically, HumanEval addresses function-level code completion, MBPP focuses on task-level code generation, DS-1000 entails statement-level code completion and insertion tasks within the data science domain, and MultiPL-E focuses on diverse programming languages code completion. This comprehensive evaluation framework ensures a thorough examination of the model's utility across various dimensions.
>
>
> > **Weakness 3**: Not clear if data leakage has been prevented. Does the evolved data or seed data overlap with HumanEval or other benchmarks' test set?
>
> To prevent data leakage, we implemented a filtering process. Similar to the method in Section 5 (Complexity and Similarity), we use all test samples as queries to retrieve the top 5 samples from training data, leveraging the SOTA embeddings model, gte-large. Subsequently, GPT-4 is employed to assess the similarity between the retrieved samples and the test sample. If a match is found, the sample is excluded. Details of the prompt and a more thorough discussion are presented in Appendix C.
>
>
> > **Weakness 4**: The paper assumes the reader to be familiar with Evol Instruct, and does not provide sufficient context to follow the method.
>
> Thank you for your suggestion. We have included the general Evol-Instruct discussion in the related work section.

---

> ### Author Response · Authors · 2023-11-20
> **Response to Reviewer 6CYQ [2/3]**
>
> > **Question 1**: Figure 3: Why are results from evol round 0 + 1 + 2 + 3 + 4 worse than 0 + 1 + 2 + 3? Do returns from Evol-Instruct start to diminish or turn negative after a certain number of rounds? This is not discussed or explored in this paper.
>
> Our empirical results reveal a potential saturation point from Code Evol-Instruct after a specific number of rounds, namely 4. This decline may be attributed to two key factors. First, continual evolution could lead to overly complicated instructions, rendering them unreasonable. Second, beyond a certain round, instructions may become excessively complex, posing a huge challenge to the evolution executing model (i.e., gpt3.5-turbo) to generate suitable responses.
>
> > **Question 2**: Analysis - Complexity and Quantity: This section is not clear to me. What are the results shown in Table 4? How does Table 4 make one conclude that the gains are not due to increase in samples or tokens? Are the 4 rows in Table 4 pass@1 metric for model checkpoints when sequentially trained on each round?
>
> In Table 4, all models are fine-tuned separately from scratch, not sequentially. We fine-tune the models with only the specific round data, which contains a similar number of samples (upper part) or tokens (lower part). The results show that with a similar number of samples or tokens, the models trained with evolved data achieve better pass@1. We have updated this section to make it clear.
>
> > **Question 3**: How dependent is the performance of Evol-Instruct on the evolving model? Will the performance substantially improve if we use GPT-4 to generate evolved rounds? Will it worsen if the evolving model is changed to other open sourced alternatives?
>
> Please refer to the section "2. Common Response to using different evolution execution models" in the Overall Response. We have provided the results of using GPT-4 or open-sourced models as evolution execution models.
>
> In additional experiments, we replaced GPT3.5 (gpt3.5-turbo) with GPT-4 for generating evolved rounds. HumanEval Pass@1 scores increased from 73.2 to 73.8 for 34B and from 59.8 to 62.2 for 15B.
>
> Furthermore, we explored using the open-source models (OSS) CodeLlama-Instruct-34B for generating evolved instructions. However, it demonstrated relatively low coding performance in response generation. To address this, we fine-tuned it using the code-alpaca dataset and utilized this model for response generation. The OSS-based Code Evol-Instruct tuned models also exhibited performance improvements compared to the code-alpaca fine-tuned model, increasing from 65.9 to 70.1 for 34B and from 45.7 to 55.5 for 15B.
>
> While GPT-4 demonstrates superior coding performance with a score of 88.4 compared to GPT3.5 (gpt3.5-turbo) with a score of 73.2, leveraging a more powerful model for generating evolved rounds does not result in a proportional performance gain (73.8 vs. 73.2). Conversely, despite CodeLlama exhibiting weaker performance than GPT3.5 (gpt3.5-turbo), the performance gap significantly diminishes when utilizing our Code-Evol Instruct to generate evolved rounds (73.2 vs. 70.1). This underscores the observation that employing a stronger model for evolved rounds can somewhat enhance performance, but the extent of improvement is not strongly correlated to the evolution executing model's coding performance. **Our Code Evol-Instruct emerges as the pivotal factor in driving performance enhancement.**

---

> ### Author Response · Authors · 2023-11-20
> **Response to Reviewer 6CYQ [3/3]**
>
> > **Question 4**: Section 3.1: We optimised the evolutionary instructions by eliminating deepening and complicating inputs, as well as In-Breadth Evolution. What does this mean? Not clear how 1 and 2 are different, can you describe precisely with an example?
>
> Thank you for your reviews. We have updated the section 3.1 to make it more clear.
>
> Example:
> Original Instruction: `Write a MongoDB query to select all documents in a collection where the field 'category' is 'clothes' and the 'brand' field is not equal to 'Nike'.`
>
> The first method "Add new constraints and requirements..." asks the model to include more requirements in the instruction: `Write a MongoDB query to select all documents in a collection where the field 'category' is 'clothes' and the 'brand' field is not equal to 'Nike', and the 'price' field is greater than or equal to 100 and less than or equal to 500.`
>
> The second method "Replace a commonly used requirement..." asks the model to replace the requirements in the instruction: `Write a MongoDB query to select all documents in a collection where the field 'material' is 'cashmere' and the 'designer' field is not equal to 'Chanel'.`
>
> > **Question 5**: Can you provide details of the compute infrastructure involved and hyperparameters in fine-tuning? (number of GPU hours, batch-size, sequence length) and other details
>
> The details of hyper-parameters are included in Section 4.2. We train our models with 24 V100 GPUs, 15 hours for 15B, and 32 hours for 34B.
>
> > **Question 6**: The abstract (and some places in the paper) comment on model sizes of Claude and Bard as 'largest closed-source LLMs', this is not verifiable, so I suggest the authors re-word this without commenting on their model size and avoid other such colloquial writing present in the paper to improve its soundness.
>
> Thank you for your suggestion. We re-word this without commenting on their model size.
>
> > **Typo** Intro Para 3: several key adaptions --> adaptations
>
> Thank you. The typo has been fixed.

---

> > ### Comment · Reviewer_6CYQ · 2023-11-22
> >
> > Dear Authors, Thank you for your detailed response and answers to the questions raised, and the results with the variants of the Evol model. I stand by my original assessment of this work. The contribution is impactful in the code domain but the approach is very similar to the one proposed in previous work in the natural language domain.

---

> > > ### Author Response · Authors · 2023-11-22
> > > **Thank You**
> > >
> > > Thank you for reviewing our paper and our responses to the questions. Your feedback is highly valuable to us, and we sincerely appreciate your positive assessment of the impact in the code domain. Your time and consideration are greatly appreciated.

---

### Author Response · Authors · 2023-11-20
**Overall Response**

Thank you for reviewing our paper! Your professional feedback provides valuable insights for crafting a more comprehensive and competitive manuscript. We appreciate that all reviewers acknowledged the idea and performance of our work. Based on the reviews, we have made updates to both the paper and its appendix. In this part, we address some common questions raised by the reviewers and the public comment.

**1. Common Response to novelty and contributions:**

> Weakness 1 of Reviewer 1: While results on code benchmarks are impressive, novelty of the scientific methodology itself is quite limited as it is an adaptation of Evol-Instruct for Code.

> Weakness 3 of Reviewer 3: While the idea is interesting, but it seems very incremental compared to prior work and the contributions are limited.

While our approach draws inspiration from the idea of previous works, **it introduces significant contributions to the Code LLMs domain**.

From the method perspective:

1. **First of Evol-based Instruction Fine-Tuning for Code**: we introduce the first evol-based instruction fine-tuning method tailored specifically to enhance the performance of code LLMs, marking a significant advancement in this field.

2. **Innovative Code-Specific Evolution Path**: Compared to previous work, we innovatively propose two new ways of evolving instructions in the code domain. First, inspired by insights from previous studies [3], we provide a valuable avenue for improving complexity of code instructions by adding constraints on time-space complexity. Second, inspired by prior research on attacking pre-trained code models [1][2], we creatively propose constructing adversarial examples by providing a piece of reference code with incorrect solutions, to help the code LLM achieve better learning performance.

3. **Evol Stop for Controlled Evolution**: Unlike the previous method, no clear mechanism for stopping evolution has been proposed. We introduce the Evol Stop mechanism that determines whether to stop the evolution process based on the performance of external dev tasks, offering a controlled and strategic approach to refining the model.

From the experiment perspective:

1. **Breakthrough Performance**: Prior to our research, open-source Code LLMs lagged behind GPT3.5(ChatGPT) by 20-30 points. Code Evol-Instruct, our novel instruction fine-tuning approach, closes this gap, marking a breakthrough for OSS Code LLMs.

2. **Novel Analysis Insights**: Our analysis introduces fresh insights into the intricate relationship between code instruction complexity (evolution) and performance. Through meticulous analysis, we explore dimensions such as evolutionary round comparisons, the correlation between quantity and complexity, and the interplay between similarity and complexity—areas not extensively explored in previous studies.

3. **Open-Sourced Models**: In contrast to closed-source models like GPT3.5(ChatGPT), our WizardCoder models are open-source, providing a potent alternative to the coding research community.

[1] Yang, Z., Shi, J., He, J., & Lo, D. (2022). Natural Attack for Pre-trained Models of Code. 2022 IEEE/ACM 44th International Conference on Software Engineering (ICSE), 1482-1493.

[2] Jha, A., & Reddy, C.K. (2022). CodeAttack: Code-based Adversarial Attacks for Pre-Trained Programming Language Models. AAAI Conference on Artificial Intelligence.

[3] Madaan, A., Shypula, A., Alon, U., Hashemi, M., Ranganathan, P., Yang, Y., Neubig, G., & Yazdanbakhsh, A. (2023). Learning Performance-Improving Code Edits. ArXiv, abs/2302.07867.

**2. Common Response to using different evolution execution models:**

Many reviewers have expressed interest in using different evolution execution models for Code Evol-Instruct. We present the results of the additional experiments here:

|  Base Model  | Evol Model | HumanEval    |
| ----- |----| ---- |
| StarCoder-15B | GPT4 | 62.2 |
| StarCoder-15B | GPT3.5(ChatGPT) | 59.8 |
| StarCoder-15B | CodeLlama | 55.5 |
| CodeLlama-34B | GPT4 | 73.8 |
| CodeLlama-34B | GPT3.5(ChatGPT) | 73.2 |
| CodeLlama-34B | CodeLlama | 70.1 |

---

### Author Response · Authors · 2023-11-21
**Overall Response: Summary of Paper Changes**

We express our gratitude to all reviewers for their thorough evaluations. We have updated the paper and its appendix as follows:

1. Updated the abstract and introduction without commenting on Claude and Bard's model size (Question 6, Reviewer 6CYQ).

2. Updated Figure 1's lower plot with different patterns to ensure better accessibility for color-blind reader (Question 4, Reviewer fiHu).

3. Updated the related work section to make a more clear discussion of general Evol-Instruct (Weakness 4, Reviewer 6CYQ).

4. Updated the related work section to include more discussions on the CodeLlama-Instruct model (Question 6, Reviewer fiHu).

5. Updated Section 3.1 to make it more clear about our Code Evol-Instruct (Question 4, Reviewer 6CYQ).

6. Updated Section 5, Evolution Models and Rounds., to include the non-GPT3.5-evol results. (Reviewer fiHu)

7. Updated Section 5, Analysis - Complexity and Quantity, to provide a clearer explanation of our analysis (Question 2, Reviewer 6CYQ).

8. Updated Appendix C to clarify our data filtering process (Weakness 3, Reviewer 6CYQ).

9. Implemented small writing improvements throughout.

We appreciate the positive feedback regarding the remarkable performance of our WizardCoder and are very excited about future work building on our model and Code Evol-Instruct!

---

### Meta-Review · Area_Chair_W58c · 2023-12-09

**Metareview:**

In this paper, the authors proposed WizardCoder, a new code LLM family fine-tuned with complex synthetic instruction-following data created following the Evol-Instruct approach. The resulting models can outperform many strong baseline models in various code generation benchmarks, including HumanEval, MBPP, DS-1000, etc.

The reviewers appreciated the idea of the instruction tuning idea with synthetically created complex data using LLMs. The reviewers also appreciated the comprehensive evaluation of the models, using open-source base models (StarCoder and CodeLLaMA) on multiple benchmarks. While there are some concerns about the novelty of the instruction data, the overall approach is quite simple yet elegant and effective in boosting the model performance significantly. There are some minor concerns about the paper, including the potential data leakage and the close relation to the original work in the NLP. I highly encourage the authors to address these comments with more detailed discussions in the final revision.

**Justification For Why Not Higher Score:**

While the overall approach is very effective in boosting the model performance significantly, the novelty of the idea is a bit limited, closely related to the Evol-Instruct approach which was originally used in the NLP domain.

**Justification For Why Not Lower Score:**

The efficacy of the proposed approach was clearly demonstrated with comprehensive and strong performance results. The paper contains good motivation and useful insights for the community to study and extend the impacts of LLMs to the code generation domain.

---

### Decision · Program_Chairs · 2024-01-16

Accept (poster)